# Vegetation changes in temperate ombrotrophic peatlands over a 35 year period

**Nicolas Pinceloup[1,2], Monique Poulin[2,3], Marie-Hélène Brice[1,2], Stéphanie Pellerin[1,2]***

**1** Institut de recherche en biologie végétale, Université de Montréal and Jardin botanique de Montréal, Montréal, Québec, Canada, **2** Québec Centre for Biodiversity Science, McGill University, Montréal, Québec, Canada, **3** Department of Phytology, Université Laval, Québec City, Québec, Canada

* stephanie.pellerin.1@umontreal.ca

**Data Availability Statement:** 1982 data are available in the Examine database of the Ministère Énergie et Ressources naturelles of Québec, (http://sigeom.mines.gouv.qc.ca/signet/classes/l1102_

## Abstract

Global changes in climate and land use are occurring at an unprecedented rate, often triggering drastic shifts in plant communities. This study aims to reconstruct the changes that occurred over 35 years in the plant communities of temperate bogs subjected to indirect human-induced disturbances. In 2015–17, we resurveyed the vascular flora of 76 plots located in 16 bogs of southern Québec (Canada) first sampled in 1982. We evaluated changes in species richness, frequency of occurrence and abundance, while considering species shade-tolerance and preferential habitat. We calculated beta diversity as between-site similarities in composition, and evaluated differences between the two surveys using tests for homogeneity in multivariate dispersion. We found a significant increase in species richness and beta diversity over the last 35 years associated with major species turnovers, indicating a biotic differentiation of the *Sphagnum*-bog plant communities. These changes were mostly associated with an increase in the abundance and frequency of shade-tolerant and facultative species, suggesting a global phenomenon of woody encroachment. Because the observed changes occurred in a few decades on sites free of *in situ* human disturbances, we suggest that they were likely induced by the synergic effect of the agricultural drainage occurring in the surrounding mineral soils, climate warming, and nitrogen atmospheric depositions. We also believe that further changes are to be expected, as the triggering factors persist. Finally, our results highlight the need for increased bog conservation or restoration efforts. Indeed, a rise in beta diversity due to the introduction of nearby terrestrial species could induce biotic homogenization of the bog flora with that of surrounding habitats and ultimately impoverish the regional species pool.

## Introduction

Global changes in climate and land use are occurring at an unprecedented rate [1,2], triggering drastic shifts in plant communities [3,4]. The overriding influence of human activities on plant communities has been shown to induce modification in species abundance [5,6] and distribution [7,8], and to increase the risk of population extinction [9,10]. At a global scale, such changes have most often been associated with biodiversity loss through the simplification

examine?l=F). Relevant data are within the manuscript and its Supporting Information files.

**Funding:** This research received financial support from the Natural Sciences and Engineering Research Council of Canada (Discovery grants to Stéphanie Pellerin Pellerin: RGPIN-2014-05367 and Monique Poulin RGPIN-2014-05663). The funders had no role in study design, data collection and analysis, decision to publish, or preparation of the manuscript.

**Competing interests:** The authors have declared that no competing interests exist.

of communities (decrease in beta diversity), a process called biotic homogenization [11,12]. Biotic homogenization usually occurs when exotic or ruderal native species replace specialist native species [11,13,14]. Nevertheless, local species invasions and extinctions are not necessarily causally related and can follow complex pathways that can also lead to biotic differentiation (increase in beta diversity) [15–17] or even no change in beta diversity [18]. Biotic differentiation has mostly been observed at local scales when the introduction of numerous exotic species outweighs the loss of natives or when disturbances increase habitat heterogeneity thus promoting the introduction of various different species [15,19–21].

Wetland flora are usually more homogeneous than terrestrial flora at local and continental scales [22,23]. This naturally low level of beta diversity within and among sites is likely related, among other reasons, to the uniformity of stressful environmental conditions found in wetlands that can be tolerated by only a small set of specialized species [22,24,25]. Despite wetlands being among the most disturbed ecosystems in the world [26], few studies have evaluated whether direct and indirect human disturbances induce further biotic homogenization in their plant communities (e.g., [17,27–29]). Among them, Price et al. [28] found a subtle process of biotic homogenization over time in the herbaceous emergent communities of palustrine wetlands of Illinois (USA). This process was associated with an increase in the presence and abundance of the exotic *Phalaris arundinacea* from the Poaceae family, and the decline of numerous other native and exotic species. On the other hand, Brice et al. [17] observed biotic differentiation in the herbaceous communities of urban riparian swamps due, in part, to the increase in habitat heterogeneity induced by urbanization. Similarly, Zhang et al. [29] showed taxonomic differentiation of macrophyte communities on a river floodplain exposed to human-induced disturbances. In fact, because many types of wetlands are naturally species-poor, they are potentially more susceptible to biotic differentiation than to homogenization following disturbances [30].

Ombrotrophic peatlands (bogs) are peat-forming naturally species-poor wetlands [31,32] that are mostly viewed as resilient ecosystems with important compositional changes usually occurring over centuries to millennia [33]. Still, several studies have shown recent rapid and drastic compositional changes in response to human-induced disturbances (drainage, eutrophication, climate warming; land-use modifications), through their influence on water and nutrient availability (e.g., [34–40]). For instance, drainage and contemporary climate change (warmer and/or drier climatic conditions) have been shown to enhance shrub and tree encroachment, hamper *Sphagnum* growth and facilitate the establishment of generalist and exotic species [33, 35,37,41]. This shift from open bog toward forested communities is also associated with a decrease in the cover and richness of other specialist bog plant species (often heliophilic species) and a concomitant increase of shade or drought-tolerant species [33,37,41–44]. More recently, a study investigating the effects of woody encroachment in bog flora, using a space for time substitution, found within-site biotic differentiation associated with the introduction of terrestrial shade-tolerant species [45]. Although vegetation changes in bogs have been the subject of numerous studies (e.g., [33;38,44]), temporal changes in plant communities at a regional scale, and how these changes affect beta diversity, remain to be explored.

In this study, we investigated the floristic changes that occurred over 35 years in the vascular plant communities of temperate bogs of southern Québec (Canada). These bogs are mostly dominated by ericaceous shrubs (e.g., *Kalmia angustifolia*, *Rhododendron groenlandicum*, *Chamaedaphne calyculata*), *Sphagnum* mosses (e.g., *Sphagnum capillifolium*, *S. magellanicum*., *S. rubellum*) and scattered *Picea mariana* or *Larix laricina* thickets. We revisited 76 plots originally surveyed in 1982 and located in 16 different bogs spread over the St. Lawrence Lowlands region. These bogs have been isolated in an agricultural landscape for more than 50 years,

which has induced a drying of the peat surface on many sites (e.g., [41,46]). We specifically aimed to 1) determine changes in taxa richness, composition and beta diversity and 2) assess whether biotic homogenization or differentiation occurred over time. We hypothesized that the studied sites would be characterized by biotic differentiation in response to species enrichment by terrestrial and shade-tolerant plants. We also expected to observe a woody encroachment phenomenon. Finally, we acknowledge that focusing on vascular plant communities may underestimate observed changes as mosses may represent a high proportion of the diversity in bogs, but mosses were rarely identified at the species level in 1982.

## Methodology

### Study area

The study area is in the St. Lawrence Lowlands, southern Québec (Canada), and extends from 45°01'N to 46°50'N latitude and from 70°55'W to 74°17'W longitude. The area is enclosed by the Canadian Shield to the north and the Appalachian Mountains to the south. The Lowlands have a flat topography and are characterized by deep arable soils derived from glacial and marine deposits. The landscape is composed of about 50% agricultural fields, 30% woodlands, 10% urbanized areas and 10% wetlands [47], of which 35% are bogs [48]. Over the last few decades, 19% of the wetland areas have been lost or disturbed by human activities, mainly agriculture and forestry, but also urbanization and peat mining [48]. Tree encroachment by *Picea mariana*, *Pinus banksiana*, *Larix laricina*, *Betula populifolia*, and *Acer rubrum* has been reported in many of the remaining bogs [35,41,45,46,49].

The mean annual temperature of the study area fluctuates from 4.1°C in the northeast to 6.7°C in the southwest [50]. Average precipitation ranges from a high of 1120 mm in the northeast to a low of 965 mm in the southwest, of which 23% and 17% fall as snow, respectively [50].

### Original sampling

During the 1980s, a large inventory campaign was conducted by the Québec Ministry of Energy and Natural Resources to evaluate the peat resource in southern Québec peatlands greater than 40 ha and with a peat deposit thicker than 30 cm [51]. During this large-scale inventory, hundreds of vegetation surveys were conducted. All plots in the study area were sampled in 1982. All plots (20 m × 20 m) were delineated in the center of a seemingly homogenous plant community. The percent cover of all vascular taxa and strata (tree, shrub, herbs, mosses) was visually estimated according to six classes: <1%, 1-5%, 6–25%, 26–50%, 51–75%, >75%. Mosses were rarely identified at the species level in original surveys and thus not investigated in the present study. Other information available in the field sheets includes geographic coordinates of the plot, peat thickness, soil pH, biotope relative dominance (pool, hummock, hollow, etc.), vegetation structure (woody, shrubby, herbaceous, etc.) and general information on the location of the plot within the bog (margins, expanse, near a drainage ditch, etc.). It should be noted that vegetation data from this original survey has never been analyzed.

### Resampling: 2015 and 2017

Historical sampling plots were not permanently marked. However, as mentioned above geographic coordinates were available from the original field sheets. Before field sampling, we discarded all original plots established on minerotrophic peatland habitats or disturbed areas. We also rejected all plots located in areas that had been affected by land conversion or subjected to woodcutting since earlier surveys. We identified these areas using 1980–2015 Google Earth

Digital Globe satellite imagery. During the summer of 2015 and 2017 (hereafter referred to as 2017 for simplification), 76 of the original plots spread over 16 bogs were relocated using a GPS (S1 Table). All plots were on private lands or on lands protected by non-governmental agencies. We obtained a verbal permission from the owner of each site before sampling. In addition to geographic coordinates, we drew upon all environmental information available from original studies in this endeavor. We surveyed the vegetation following the same methodology used in 1982. Due to the abundance of visual landmarks, we estimate that we were able to relocate all plots within 30–40 m from their original position. Although a relocation error could introduce a source of variance in the dataset, systematic biases favoring vegetation change in one direction are unlikely to occur, especially when environmental variables are used in combination with geographic location to relocate unmarked plots [13,52].

## Data analysis

Before analyses, taxonomic nomenclature was standardized according to the Database of Vascular Plants of Canada (VASCAN) [53]. We lumped all sedge taxa (*Carex*) for both years at the genus level, as very few individuals were identified to the species level in 1982. We also merged *Vaccinium myrtilloides* and *Vaccinium angustifolium* under *Vaccinum* cf. *angustifolium* because they were often undistinguished in 1982.

**Taxa richness.** Change in taxa richness at the plot level (alpha diversity) between 1982 and 2017 was evaluated using a Wilcoxon signed rank test. We also considered each taxa's habitat preferences by comparing richness of taxa sorted by their wetland indicator status [54]. Accordingly, taxa that almost always occur in wetlands were classified as "obligate", those that usually occur in wetlands, but may occur in non-wetland habitats as "facultative", and those that occur equally in wetlands and non-wetland habitats or that usually occur in non-wetland habitats as "non-wetland" (S2 Table). No taxa that almost always occur in non-wetland habitats (upland) were found. The wetland indicator status of each taxon was retrieved from the National Wetland Plant List [55] and Lapointe et al. [56]. Generalized Poisson mixed models were used to compare the number of taxa between groups of habitat preferences and between years as well as the interaction between both factors. Plots were treated as a random factor, allowing us to control for pairing between plots and dependence between the numbers of taxa of different preferences in the same plot. Because a significant interaction was found between year and habitat preference, we conducted post-hoc Tukey tests between habitats independently for each year. We verified that model residuals followed a normal distribution of homogeneous variance.

We also compared richness of taxa according to their tolerance to shade [57] (S2 Table). We classified taxa into three broad categories of shade tolerance: "tolerant" for taxa that grow well under a limited amount of light, "mid-tolerant" for taxa that can tolerate shade during some phases of their life cycle, and "intolerant" for taxa that only grow with a high amount of available light. Shade tolerance information was retrieved from the online databases TRY [58] and PLANTS [59], as well as from Humbert et al. [60]. We used the same linear mixed-effect model procedure described above.

**Taxa frequency and abundance.** Taxa with the greatest changes in terms of frequency of occurrence were identified by comparing the number of plots occupied by each taxon for each year using Chi-square goodness-of-fit tests (taxa were tested one by one). A Yates continuity correction was applied [61]. Finally, we evaluated whether the abundance of some taxa changed over time using plant cover. The cover data were transformed into percentages using the midpoint of each class. Only species with a mean of $\geq 5\%$ coverage for at least one of the sampling periods were used (12 species analyzed). Differences between 1982 and 2017 were tested using a two-tailed paired t-test (species were tested one by one).

**Beta diversity.** Differences in beta diversity between 1982 and 2017 were analyzed using a distance-based test for homogeneity of multivariate dispersions (PERMDISP) [62]. PERM-DISP calculates the distance of each site (here, plot) to the centroid in an ordination space (principal coordinate analysis) and then tests whether these distances are different between groups (here, sampling years) through permutation tests. More precisely, two site-by-species matrices were first computed; one using covers (median of cover classes) and the other using presence-absence data. Then, for each matrix, a site-by-site distance matrix was built using the Hellinger distance [63]. This distance matrix was used to compute the centroid of each group of sites. The distance of each site to its associated group centroid was calculated, and the dispersion of these distances (within-group variance) was used as an estimate of beta diversity (the greater the within-group variance, the higher the beta diversity). The site distances to centroid were subjected to a paired t-test with 9999 permutations to determine whether dispersions differed between groups. To detect a shift in species composition between year (i.e., turnover), we tested for location differences between centroids using PERMANOVA with pseudo-F ratios (9999 permutations) [64]. Because this test is sensitive to differences in multivariate dispersions [65], data visualization was used to support the interpretation of the statistical test. The differences in multivariate dispersion and composition were illustrated in principal coordinates analysis ordinations (PCoA). Finally, we decomposed beta diversity into replacement (turnover) and richness difference (species gain/loss) using Jaccard dissimilarity indices and presence-absence data [66]. We decomposed beta diversity for each year as well as between years.

All statistical analyses were performed using R 3.3.2 [67]. Wilcoxon signed rank tests were performed using the *wilcox.test*, chi-square tests using *chisq.test*, paired t-test using *t.test*, and generalized Poisson mixed models using glm(), all from the *stats* package [67]. Tukey tests were done with the *lsmeans()* function in the *lsmeans* library [68]. Hellinger transformations were done using *decostand()*, multivariate dispersion analyses were performed using *betadisper()*, centroid locations were tested using *adonis2()*, all from the *vegan* package [69]. Permutation t-tests were done using *perm.test()*, from the *broman* package [70]. Beta diversity partitioning was conducted using the *beta.div.comp* function [66].

## Results

### Taxa richness

A total of 37 vascular taxa were recorded in 1982 and 57 in 2017 (total = 67), and only one exotic (*Frangula alnus*) was found, in 2017 (S2 Table). Thirty new taxa were detected in 2017, while ten were lost. Most of the taxa gained were found in <10 plots, except *Aronia melanocarpa* (33 plots), *Gaylussacia baccata* (18 plots), *Vaccinium corymbosum* (18 plots), *Eriophorum virginicum* (17 plots), *Platanthera blephariglottis* var. *blephariglottis* (14 plots) and *Coptis trifolia* (11 plots). Furthermore, few of the taxa gained were abundant in the plots where they were found (mean cover >5%): *Rubus allegheniensis* (39% mean cover; 4 plots), *V. corymbosum* (24%; 18 plots), *Frangula alnus* (22%; 3 plots), *G. baccata* (12%; 18 plots) and *Osmundastrum cinnamomeum* (8%; 5 plots). All taxa lost were present in five plots or less in 1982, except *Alnus incana* subsp. *rugosa* (8 plots).

Taxa richness per plot ranged from 3 to 15 in 1982 and from 7 to 21 in 2017 (Fig 1). The number of taxa increased in 61 plots, remained similar in five plots and decreased in ten plots. Overall, mean richness per plot significantly increased over time, from an average of 10 to 14 taxa (V = 165; $p \leq 0.005$).

The number of "facultative" and "non-wetland" taxa per plot significantly increased with time, while the number of "obligate" taxa remained similar (Fig 2A). The number of "obligate"

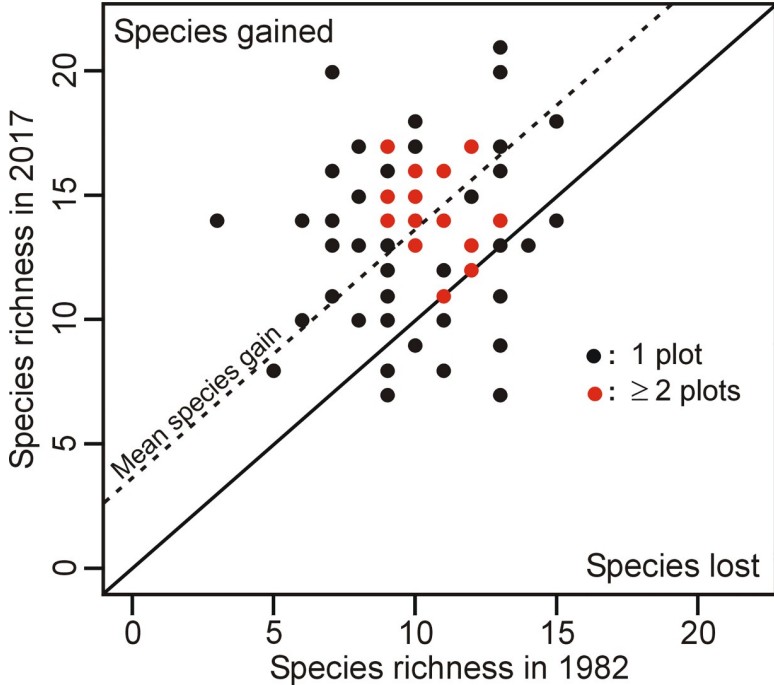

**Fig 1. Vascular taxa richness in temperate bogs of southern Québec (Canada) in 1982 and 2017.** Plots sampled in 1982 are compared to the same plots re-sampled in 2017. The plain diagonal line is the 1:1 line (no change in richness between the two years) and the dashed line represents the mean richness increase. Grey dots represent at least two plots with the exact same number of species in 1982 and 2017.

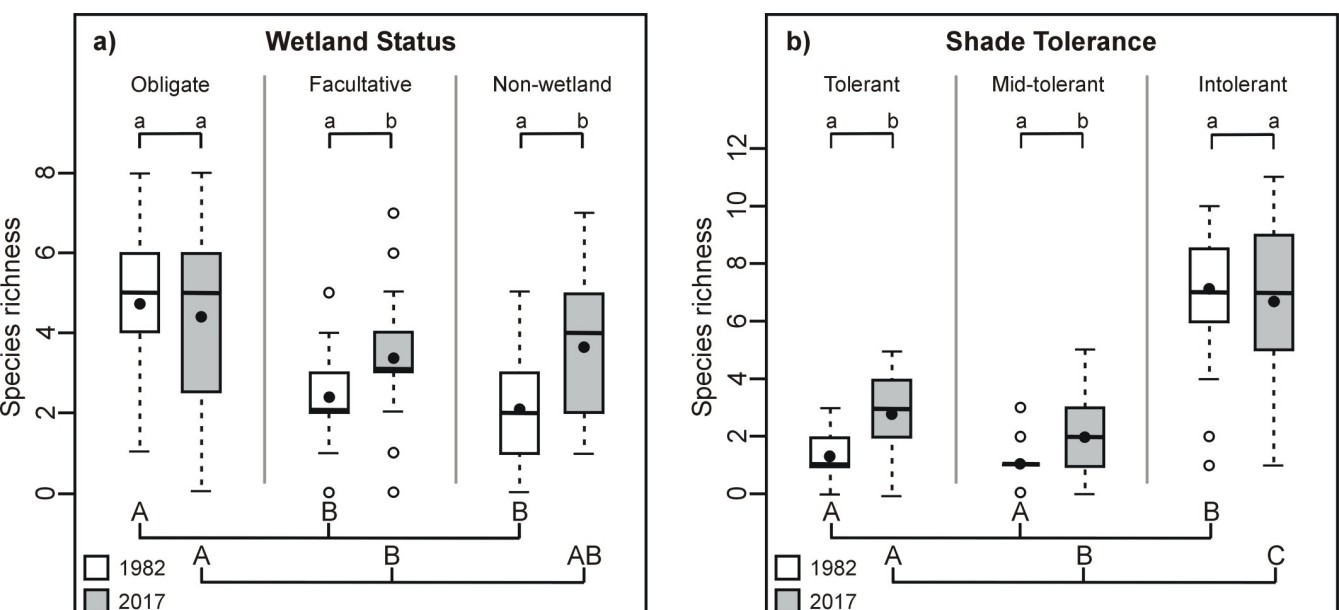

**Fig 2.** Richness of a) obligate, facultative and non-wetland taxa and b) shade tolerant, mid-tolerant and intolerant taxa in 1982 and 2017 in temperate bogs of southern Québec (Canada). Provided are mean taxa richness (black dot), median (line), 25–75% quartiles (boxes) and ranges (whiskers). Different lower-case letters indicate a significant difference (α = 0.05; Generalized Poisson mixed models) between years, whereas different upper-case letters indicate a significant difference (α = 0.05; Tukey's test) within the year.

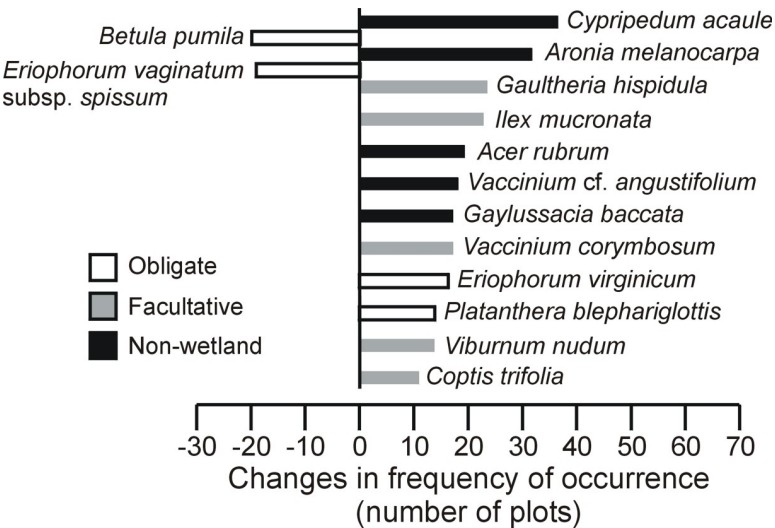

**Fig 3. Plant taxa with significant changes ($p \leq 0.05$) in the frequency of occurrence (number of plots where species occurs) between 1982 and 2017 in temperate bogs of southern Québec (Canada).** Taxa were tested one by one (chi-square goodness-of-fit tests with Yates correction). On the left side are taxa with lower frequency in 2017 than in 1982, and on the right side, taxa with higher frequency in 2017 than in 1982.

taxa was about two times greater than that of "facultative" and "non-wetland" taxa in 1982 and about 1.5 times greater in than that of "facultative" taxa in 2017 (Fig 2A). The amount of shade "tolerant" and "mid-tolerant" taxa per plot significantly increased over time, while the richness of shade "intolerant" taxa remained similar for both years (Fig 2B). It follows that the number of "shade-intolerant" taxa was greater than that of the two other groups of taxa for both years.

## Species frequency and abundance

Of the 67 taxa tested, 14 significantly changed their frequency of occurrence over time (Fig 3) and six experienced a significant change in their abundance (Fig 4). More precisely, 11 taxa significantly increased in frequency, among which *Acer rubrum* and *Vaccinium corymbosum* also increased in abundance. The increase in frequency was particularly marked ($\geq$20 plots) for *Cypripedium acaule*, *Aronia melanocarpa*, *Gaultheria hispidula*, *Ilex mucronata* and *Acer rubrum*. Taxa that became more frequent or abundant over time were mostly "facultative" or "non-wetland" taxa. Inversely, two taxa significantly decreased in frequency (Fig 3) and four decreased in abundance (Fig 4). Only *Eriophorum vaginatum* subsp. *spissum* decreased both in frequency and abundance. While *E. vaginatum* was still found in 43 plots in 2017 (62 plots in 1982), *Betula pumila* was found in only one plot the same year (22 plots in 1982). Taxa decreasing in frequency or abundance were all wetland "obligate" taxa.

## Beta diversity

Beta diversity increased between 1982 and 2017 (presence-absence: T = 98.79; $p< 0.0001$; abundance data: T = 109.28; $p< 0.0001$), meaning that vascular plant communities of bogs differed more markedly from one site to another in 2017 than in 1982 (Fig 5). This increase in beta diversity was associated with composition turnover as a significant difference in centroid position between years was also found for both presence-absence (F = 9.085; $p\leq 0.0001$) and abundance data (F = 9.514; $p\leq 0.0001$).

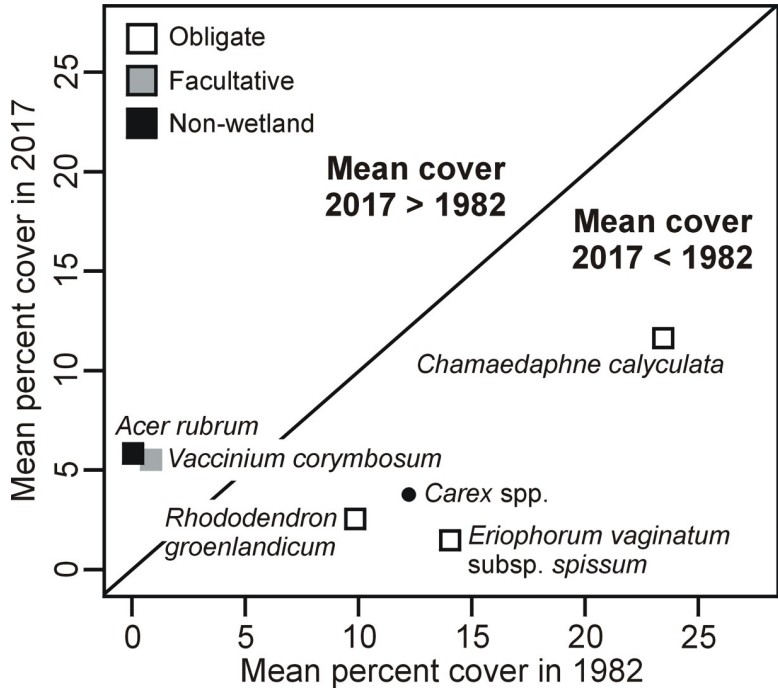

**Fig 4. Plant taxa mean cover in 2017 vs 1982 in temperate bogs of southern Québec (Canada).** Only taxa with significant change in mean cover (two-tailed paired t-test; $p \leq 0.05$) are presented. The plain line is the 1:1 line. *Carex* spp. is plotted with a different symbol, as all species were lumped together and, therefore, not associated to a particular wetland status.

Finally, beta diversity in both 1982 and 2017, as well as beta diversity changes between 1982 and 2017, resulted mostly from taxa replacement rather than richness differences (Fig 6). For instance, when comparing all pairs of plots between 1982 and 2017, replacement accounted for 48.5% of the beta diversity decomposition while richness difference accounted only for 12.6%.

## Discussion

In this study, we hypothesized that the studied sites would be characterized by a temporal biotic differentiation of the vascular flora in temperate bogs in response to species enrichment by terrestrial shade-tolerant plants. Our results showed that the richness of the studied bogs increased from 1982 to 2017, mostly due to the establishment of facultative and shade-tolerant plants. As expected, these changes induced a biotic differentiation, since we found higher beta diversity in 2017 than 35 years prior. This decrease in similarity with time mostly represented the outcome of species turnover. This indicates that newly established taxa are not similar across the study area and/or that various taxa are being replaced over time.

### Changes in plant environmental preferences

The analyses of plant richness according to individual habitat preference indicated an enrichment of taxa associated to drier habitats as we found more facultative and non-wetland plants in recent surveys than in the older ones. For example, *Acer rubrum*, a species that usually occur in non-wetland habitats, which was present in only five plots in 1982, is more recently present in 25 plots and has experienced a significant increase in cover over time (Figs 3 and 4). *Acer rubrum* introduction and increase in density, often following drainage, have been recently documented in many North American wetlands including bogs [50,71,72]. On the other hand,

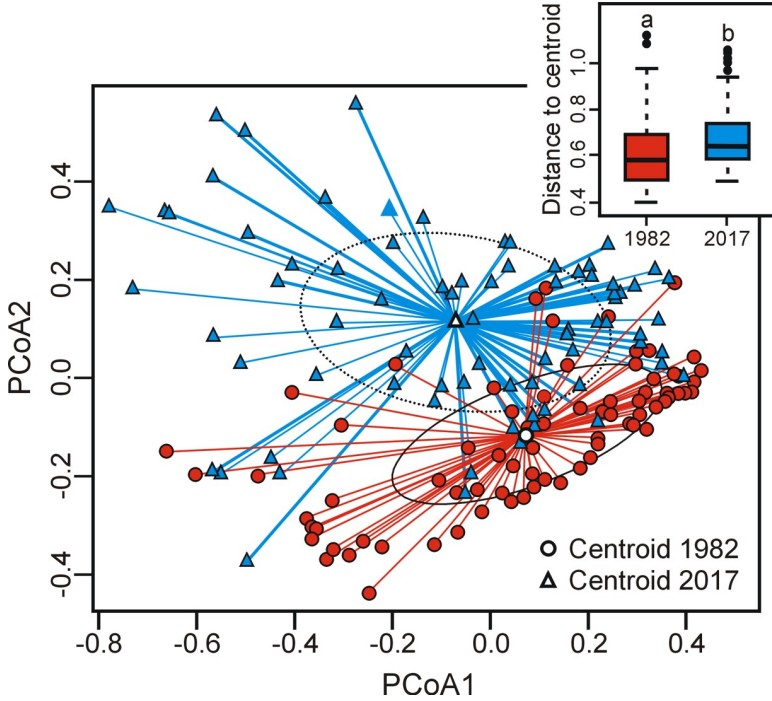

**Fig 5. Effect of time on beta diversity for vascular plant taxa in bogs of southern Quebec (Canada), using presence-absence data.** Beta diversity was measured as the distance of sites to their group centroid, here represented on the first two axes of the PCoA and using the boxplot of the sites-to-centroid distance. Provided are median (line), 25–75% quartiles (boxes) and ranges (whiskers). Different letters indicate a significant difference ($p < 0.05$) determined by t-test. Circles are ellipses of standard deviation.

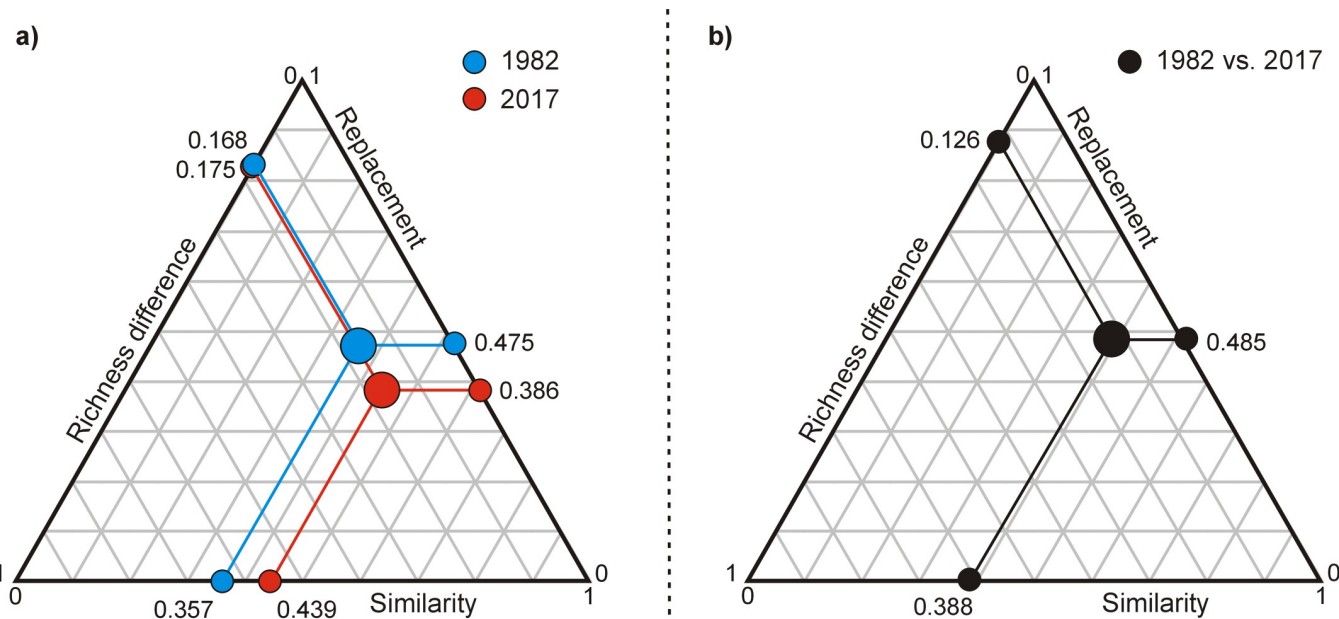

**Fig 6.** Triangular plot of the relationship among a) the 76 plots in 1982 and 2017, and b) for all pairs of plots between 1982 vs 2017, illustrating the contributions of three components of beta diversity over time: similarity of species composition, richness difference and replacement of species. The large dots inside the triangle represent the centroid, and the smaller dots represent the mean values of the three components. Values are indicated as raw percentages (i.e. 0.168 equals 16.8% of total beta diversity).

although the richness of obligate wetland plants did not significantly change over time, some of these plants experienced a noteworthy decrease in frequency and abundance. This was the case for *Eriophorum vaginatum* subsp. *spissum*, a deep rooting wetland obligate typical of open bogs [73]. Despite the strong capacity of *E. vaginatum* to persist in peatlands, notably due to its typical tussock growth with basal meristematic tissue, surveys in vacuum mined peatlands have revealed its sensitivity to water-table drawdown [74].

Our results suggest that the studied bogs experienced an increase in shading level over time, as indicated by the gain of shade-tolerant and mid-tolerant plants. This increased level of shading is likely the result of shrub and tree encroachment in bogs, as suggested, among other reasons, by the increase in frequency and abundance of *Acer rubrum* and *Vaccinium corymbosum*. In fact, a quarter of the plots investigated presented an increase in tree cover (mean increase of 4.7%) and, more importantly, three quarter of the plots showed an increase in shrub (mean increase of 37.2%) cover (S3 Table). While our results indicate that shaded habitats are becoming more widespread in the studied bogs, we also found no significant change in the richness of shade-intolerant species, although the variability of the data was higher in 2017 than in 1982 (Fig 1). Furthermore, most of the species that significantly decreased in occurrence and/or abundance (*Betula pumila*, *Chamaedaphne calyculata*, *E. vaginatum*) were shade-intolerant. The lack of significant change in the richness of the shade-intolerant species at the plot level may be due to the time-lag after disturbance, which implies that changes may still occur in the coming decades [75]. Overall, a more variable hydrologic environment and structure of plant strata likely allowed the co-occurrence of heliophilic bog specialists and shade and drought-tolerant plants more typical of forest environments. These results concur with the phenomenon of woody encroachment increasingly observed in bogs worldwide (e.g., [35,40,41,50]).

## Beta diversity

As expected, we found a biotic differentiation (increase in beta diversity) over a period of 35 years in temperate *Sphagnum*-dominated bogs of the southern Québec. Changes in beta diversity are led by two primary components: species turnover and richness differences [30,66]. Although we found an increase in plant richness over time in nearly all the sampling plots, our results indicated that the observed changes in beta diversity were mainly due to species replacement. For instance, our beta diversity partitioning analyses indicated that the influence of species replacement was four times greater than that of richness difference in beta diversity changes between 1982 and 2017. A recent study using a time-for-space substitution, conducted in the same study area but in different bogs, also found a higher beta diversity on sites recently subjected to woody encroachment than in open bog habitats, and this increase was also mostly driven by species turnover [45]. Significant species turnover with time, mainly driven by the replacement of typical bog species by trees, shrubs and other widespread forest species, was also found in Sweden [33,37], although the effect on beta diversity was not analysed.

## Potential causes of the observed changes

Rapid changes in peatland plant communities have been mostly attributed to human-induced disturbances such as increasing temperature, atmospheric nitrogen deposition and drainage [34–40, 76]. In the studied bogs, the observed changes probably resulted from the interaction among all these factors, of which those influencing the hydrology of the sites likely predominated. Because the study sites are located in seemingly undisturbed and undrained bogs, the gradual drying of the peat deposits likely resulted from the extensive drainage of mineral soils in the surrounding agricultural catchments [41,46,50]. For instance, using paleohydrological

reconstruction (testate amoebae), it was recently showed that the water level drawdown in two bogs of the study area matches the peak of agricultural activities and ditch-digging mechanization in their surrounding mineral catchment [46]. On the other hand, since 1960, southern Québec has experienced an average temperature increase of 1.2˚C [77], which may also have contributed to peat drying. Indeed, according to a modeling study [78], even a 1˚C temperature increase can dry the peat surface enough to shift open bogs into tree-dominated ecosystems. In conjunction with regional drainage and climate warming, nutrient enrichment may have facilitated the observed changes. Indeed, six of the 16 bogs studied are located in a region with some of the highest levels of wet nitrogen deposition in eastern North America [79]. All the bogs might also have been enriched by dry mineral dust from adjacent agricultural lands [39,46]. Finally, we could not rule out that some of the changes may also be associated with intrinsic ecosystem dynamics such as peat accumulation and mostly competition among plants (e.g., [80–82]).

## Conclusion

In conclusion, our results reveal that rather rapid compositional changes in the plant communities of temperate *Sphagnum*-dominated bogs isolated in an agricultural landscape induced an overall biotic differentiation. Because these changes occurred in a few decades within sites free of *in situ* human disturbances, we suggest that they were likely induced by the synergy effect of drainage in the surrounding mineral agricultural soils, climate warming, and nitrogen atmospheric depositions, as observed in several other bogs. We also believe that further changes are to be expected, since the triggering factors persist and obligate and shade-intolerant species remain abundant, unlike other bogs, where woody encroachment is more pronounced (e.g., [35]).

Finally, our results also highlight the need for increased conservation efforts in such habitats, as bogs face two challenges: their surface is being lost to agriculture and urbanization [48] and their floristic distinctiveness is disappearing. Indeed, peatlands are usually species poor ecosystems with a highly specialized flora. An increase in beta diversity of their plant community due to the introduction of nearby species could increase their similarity with surrounding mesic habitats and ultimately impoverish the regional species pool.

## Supporting information

**S1 Table. Geographical coordinates (degree decimal) of the plots.**
(DOCX)

**S2 Table. List of plant taxa sampled.** Taxa name, shade-tolerance (T = tolerant; M = mid-tolerant; I = intolerant), wetland indicator status (Obl = obligate; Fac = facultative; NW = non-wetland), frequency of occurrence for both years (number of plots) and in brackets mean abundance in the plots where the species is present are indicated. NA indicates inconsistent or unavailable information.
(DOCX)

**S3 Table. Tree and shrub cover (in %) in sampled plots for both survey periods.**
(DOCX)

## Acknowledgments

We are grateful to the landowners who allowed us to work on their lands, to field assistants, to S. Daigle and P. Legendre for statistical advice and K. Grislis for linguistic revision.

## Author Contributions

**Conceptualization:** Nicolas Pinceloup, Monique Poulin, Stéphanie Pellerin.

**Data curation:** Nicolas Pinceloup, Marie-Hélène Brice, Stéphanie Pellerin.

**Formal analysis:** Nicolas Pinceloup, Marie-Hélène Brice.

**Funding acquisition:** Monique Poulin, Stéphanie Pellerin.

**Methodology:** Nicolas Pinceloup, Monique Poulin, Marie-Hélène Brice, Stéphanie Pellerin.

**Resources:** Monique Poulin, Stéphanie Pellerin.

**Validation:** Stéphanie Pellerin.

**Writing – original draft:** Nicolas Pinceloup, Monique Poulin, Marie-Hélène Brice, Stéphanie Pellerin.

**Writing – review & editing:** Nicolas Pinceloup, Monique Poulin, Stéphanie Pellerin.

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
