## [Decision Letter · Decision Letter 0]

16 Dec 2019

PONE-D-19-25600

Vegetation changes in temperate ombrotrophic peatlands over a 35 year period

PLOS ONE

Dear Dr Pellerin,

Thank you for submitting your manuscript to PLOS ONE. After careful consideration, we feel that it has merit but does not fully meet PLOS ONE’s publication criteria as it currently stands. Therefore, we invite you to submit a revised version of the manuscript that addresses the points raised during the review process.

ACADEMIC EDITOR:

Please carefully consider the comments of the reviewer and revise the manuscript accordingly.In the revising process,you should clarify the methods used and make the discussions more focus on the major findings of the study.

We would appreciate receiving your revised manuscript by Jan 30 2020 11:59PM. To enhance the reproducibility of your results, we recommend that if applicable you deposit your laboratory protocols in protocols.io, where a protocol can be assigned its own identifier (DOI) such that it can be cited independently in the future. For instructions see: http://journals.plos.org/plosone/s/submission-guidelines#loc-laboratory-protocols

We look forward to receiving your revised manuscript.

Kind regards,

RunGuo Zang

Academic Editor

PLOS ONE

Journal Requirements:

**When submitting your revision, we need you to address these additional requirements**:

**Please ensure that your manuscript meets PLOS ONE's style requirements, including those for file naming**. **The PLOS ONE style templates can be found at http://www.plosone.org/attachments/PLOSOne_formatting_sample_main_body.pdf and http://www.plosone.org/attachments/PLOSOne_formatting_sample_title_authors_affiliations.pdf**In your Methods section, please provide additional location information of the sampling sites, including geographic coordinates for the data set if available.In your Methods section, please provide additional information regarding the permits you obtained for the work. Please ensure you have included the full name of the authority that approved the field site access and, if no permits were required, a brief statement explaining why.We note that [Figure1 ] in your submission contain [map/satellite] images which may be copyrighted. All PLOS content is published under the Creative Commons Attribution License (CC BY 4.0), which means that the manuscript, images, and Supporting Information files will be freely available online, and any third party is permitted to access, download, copy, distribute, and use these materials in any way, even commercially, with proper attribution. For these reasons, we cannot publish previously copyrighted maps or satellite images created using proprietary data, such as Google software (Google Maps, Street View, and Earth). For more information, see our copyright guidelines: http://journals.plos.org/plosone/s/licenses-and-copyright.

You may seek permission from the original copyright holder of Figure(s) [#] to publish the content specifically under the CC BY 4.0 license.

If you are unable to obtain permission from the original copyright holder to publish these figures under the CC BY 4.0 license or if the copyright holder’s requirements are incompatible with the CC BY 4.0 license, please either i) remove the figure or ii) supply a replacement figure that complies with the CC BY 4.0 license. Please check copyright information on all replacement figures and update the figure caption with source information. If applicable, please specify in the figure caption text when a figure is similar but not identical to the original image and is therefore for illustrative purposes only.

5. Please upload a copy of Supporting Information S1 Table and S2 Table, which you refer to in your text on page 25.

Additional Editor Comments (if provided):

Please carefully consider the concerns of the reviewer,and revise the manuscript accordingly

Reviewers' comments:

Reviewer's Responses to Questions

**Comments to the Author**

1. Is the manuscript technically sound, and do the data support the conclusions?

Reviewer #1: Yes

2. Has the statistical analysis been performed appropriately and rigorously? 

Reviewer #1: Yes

3. Have the authors made all data underlying the findings in their manuscript fully available?

Reviewer #1: Yes

4. Is the manuscript presented in an intelligible fashion and written in standard English?

Reviewer #1: Yes

5. Review Comments to the Author

Reviewer #1: In this paper Authors tested the temporal change of community composition and structure over 35 years in Sphagnum-dominated bogs isolated in an agricultural landscape. I sincerely liked reading this manuscript, it is well writing respecting scientific standards and easy to follow. This paper has as great potential and I recommend it to publication. I want to thank authors for the clarity of the methods and result section, it was very easy to follow. The methods are very well design to answered to questions. I especially appreciated the use of different life history trait to disentangles the mechanism underling the temporal change of bogs plant community. In the same way, authors use both presence-absence and abundance in their analysis of beta diversity, which is very useful to identify different mechanisms. However, I have a concern about the statistical methods use (see my comment on Methods section about the richness distribution). Finally, the discussion wrap-up all results and introduce them in the more general scientific context. This is a high-quality piece of work.

Abstract.

You interpret the dynamic of plant community by the synergy between drainage, climate warming, and nitrogen atmospheric depositions. I wonder why did not directly address the effect of intrinsic ecosystem dynamic. Especially with sphagnum bogs, naturally dedicated to evolving toward a woody state. (You did explain this hypothesis in the introduction L74-75).

The last sentence is very complex and mixed many ideas not directly introduce previously in the abstract. Thus, it is hard to follow. I truly believe this last sentence could be divided and that the conservation proposition better introduces.

Introduction

L68: “with compositional changes usually occurring over long time scales”. Could you give an idea how long is "long time"? For instance, depending on of the field study, 35 years could either be long (meadow, bog?) or short (forest) term in the temporal dynamic of plant community.

L 82-83: Since your study system is the “Sphagnum-dominated bogs” and that your research question focus on vascular plant could you give more information about the kind of community we are talking about. It is quite easy for a bryologist to figure out what kind of ecosystem is about, but I think harder for a general ecologist/plant ecologist.

L 87-91: Following the previous comment, focusing on the temporal dynamic of taxonomic diversity and composition of vascular plant communities in Sphagnum-dominated bogs, meaning peatland ecosystem characterize by high diversity of bryophytes, would not aim at forgetting a considerable part of biodiversity? You gave the answered to my comments L118-119 of the Introduction, but I would have liked to read it earlier.

Methods.

L137-140: This is a very important statement, that I support by principle.

L151-154: I truly believed that it could be much easier for the reader (meaning, not necessary a specialist of the question) to follow if you attribute different name to species occurring sometimes in wetland (facultative wetland) and species occurring most of the time outside of wetland (facultative) with different names.

L157-158: I ask author to verify the validity of the statistical analysis comparing the number of taxa between groups of habitat preferences. Since they analyzed computing, the most probable distribution is Poisson then the indicating analysis is GLM with family link Poisson. Usually, both analyses give similar results, but please could you verify with the correct distribution parameter? You can keep the exact same model structure.

L164: I wonder if the distribution was normal given the nature of raw data (=counting)!

L183-189: This is a very good procedure; it ensures to disentangle different mechanism acting at the community composition (which species) and structure level (dominant vs rare species).

L231: Fig. 2. The last sentence of the caption is not very explicit: “Grey dots represent more than one plot” I recommend this slightly different sentence: Grey dots represent species occurring in more than one plot.

Fig. 4. Please add: “number of plots where specie occurs” in the bracket.

Fig 7. The legend is not easy to read, please make the line of the legend and the figure thicker.

6. PLOS authors have the option to publish the peer review history of their article (what does this mean?). If published, this will include your full peer review and any attached files.

Reviewer #1: No

---

## [Author Response · Author response to Decision Letter 0]

29 Jan 2020

Answer. Done.

2. In your Methods section, please provide additional location information of the sampling sites, including geographic coordinates for the data set if available.

Answer. We added a table given all the geographic coordinates of the sampling plots as supplementary material (S1 Table). We also indicated that all sampling sites were on private lands or on land protected by non-governmental agencies. L137-139

Answer. No permit was needed. A sentence was added to the method section: “All plots were on private lands or on lands protected by non-governmental agencies. We obtained a verbal permission from the owners of each site for sampling.” L137-139.

4. We note that [Figure1 ] in your submission contain [map/satellite] images which may be copyrighted. All PLOS content is published under the Creative Commons Attribution License (CC BY 4.0), which means that the manuscript, images, and Supporting Information files will be freely available online, and any third party is permitted to access, download, copy, distribute, and use these materials in any way, even commercially, with proper attribution. For these reasons, we cannot publish previously copyrighted maps or satellite images created using proprietary data, such as Google software (Google Maps, Street View, and Earth). 

Answer. We removed the figure and instead added a table given all the geographic coordinates of the sampling plots as supplementary material (S1 Table)

5. Please upload a copy of Supporting Information S1 Table and S2 Table, which you refer to in your text on page 25.

Answer. Done

Reviewer 1:

In this paper Authors tested the temporal change of community composition and structure over 35 years in Sphagnum-dominated bogs isolated in an agricultural landscape. I sincerely liked reading this manuscript, it is well writing respecting scientific standards and easy to follow. This paper has as great potential and I recommend it to publication. I want to thank authors for the clarity of the methods and result section, it was very easy to follow. The methods are very well design to answered to questions. I especially appreciated the use of different life history trait to disentangles the mechanism underling the temporal change of bogs plant community. In the same way, authors use both presence-absence and abundance in their analysis of beta diversity, which is very useful to identify different mechanisms. However, I have a concern about the statistical methods use. Finally, the discussion wrap-up all results and introduce them in the more general scientific context. This is a high-quality piece of work.

Answer. Thank you for this positive review and kind recognition of the efforts put into this work. We performed new statistical analyses as recommended (see answer to comments below). 

Abstract

You interpret the dynamic of plant community by the synergy between drainage, climate warming, and nitrogen atmospheric depositions. I wonder why did not directly address the effect of intrinsic ecosystem dynamic. Especially with sphagnum bogs, naturally dedicated to evolving toward a woody state. (You did explain this hypothesis in the introduction L74-75).

Answer. The view that Sphagnum bogs are always naturally evolving toward a woody state have been largely criticized (e.g., Walker, 1970 Klinger et al. 1990; Klinger 1996). Bogs can, however, naturally alternate from open to forested states and vice versa, mostly in response to long-term climate changes (Birks 1975; Gear & Huntley 1991; Pilcher et al. 1995; Penda 2011; Heijmans et al., 2013). In the study area a large number of sites are affected by woody encroachment despite the fact that bog inception in these sites did not occur synchronously (e.g. Pellerin & Lavoie 2003; Lavoie et al., 2013; Beauregard et al. 2019). This suggested that intrinsic ecosystem dynamic is not the main cause of woody encroachment in the area. We did not modify the abstract, but we added a sentence on this aspect in the discussion: “Finally, we could not rule out that some of the changes may also be associated with intrinsic ecosystem dynamics such as peat accumulation and mostly competition among plants [e.g., 80-82].” L370-372

The last sentence is very complex and mixed many ideas not directly introduce previously in the abstract. Thus, it is hard to follow. I truly believe this last sentence could be divided and that the conservation proposition better introduces.

Answer. We agree with the reviewer. This sentence was split in two and we modified the wording to simplify our idea. “Finally, our results highlight the need for increased bog conservation or restauration efforts. Indeed, a rise in beta diversity due to the introduction of nearby terrestrial species could induce biotic homogenization of the bog flora with that of surrounding habitats and ultimately impoverish the regional species pool.” L28-31

Introduction

L68: “with compositional changes usually occurring over long time scales”. Could you give an idea how long is "long time"? For instance, depending on of the field study, 35 years could either be long (meadow, bog?) or short (forest) term in the temporal dynamic of plant community.

Answer. We changed “long term” by centuries to millennia. L67

L 82-83: Since your study system is the “Sphagnum-dominated bogs” and that your research question focus on vascular plant could you give more information about the kind of community we are talking about. It is quite easy for a bryologist to figure out what kind of ecosystem is about, but I think harder for a general ecologist/plant ecologist.

Answer. We now talk about open bogs instead of Sphagnum-dominated bogs as we found it repetitive. We added a summary description of the bogs in the study area: “These bogs are mostly dominated by ericaceous shrubs (e.g., Kalmia angustifolia, Rhododendron groenlandicum, Chamaedaphne calyculata), Sphagnum mosses (e.g., S. capillifolium, Sphagnum magellanicum., S. rubellum) and scattered Picea mariana or Larix laricina thickets.” L83-86. 

L 87-91: Following the previous comment, focusing on the temporal dynamic of taxonomic diversity and composition of vascular plant communities in Sphagnum-dominated bogs, meaning peatland ecosystem characterize by high diversity of bryophytes, would not aim at forgetting a considerable part of biodiversity? You gave the answered to my comments L118-119 of the Introduction, but I would have liked to read it earlier.

Answer. Accordingly, we added a sentence at the end of the Introduction to explained why we analysed vascular plant communities only: “Finally, we acknowledge that focusing on vascular plant communities may underestimate observed changes as mosses may represent a high proportion of the diversity in bogs, but mosses were rarely identified at the species level in 1982.” L94-97

Methods.

L137-140: This is a very important statement, that I support by principle.

Answer. Thanks.

L151-154: I truly believed that it could be much easier for the reader (meaning, not necessary a specialist of the question) to follow if you attribute different name to species occurring sometimes in wetland (facultative wetland) and species occurring most of the time outside of wetland (facultative) with different names.

Answer. Accordingly, we changed throughout the manuscript the name of the species occurring most of the time outside of wetland for « non-wetland » and those occurring sometimes in wetland for “facultative”.

L157-158: I ask author to verify the validity of the statistical analysis comparing the number of taxa between groups of habitat preferences. Since they analyzed computing, the most probable distribution is Poisson then the indicating analysis is GLM with family link Poisson. Usually, both analyses give similar results, but please could you verify with the correct distribution parameter? You can keep the exact same model structure.

Answer. The reviewer was wright; we redid our analyses using Generalized Poisson mixed models (L162-169). The results remain the nearly same, we changed the Results section accordingly (L238-241). 

L164: I wonder if the distribution was normal given the nature of raw data (=counting)!

Answer. Following the Poisson transformation the residuals followed a normal distribution of homogeneous variance (L168-169).

L183-189: This is a very good procedure; it ensures to disentangle different mechanism acting at the community composition (which species) and structure level (dominant vs rare species).

Answer. We are glad that the methods are approved by the reviewer.

L231: Fig. 2. The last sentence of the caption is not very explicit: “Grey dots represent more than one plot” I recommend this slightly different sentence: Grey dots represent species occurring in more than one plot.

Answer. In this figure (now Figure 1) each dot represents a site with a specific number of species in 1982 and 2015. As 2 sites may have exactly the same number of species in both years dot of these two plots superimposed. We clarified this aspect in the figure caption. “Grey dots represent at least two plots with the exact same number of species in 1982 and 2017.” L 236-237.

Fig. 4. Please add: “number of plots where specie occurs” in the bracket.

Answer. Done (now Figure 3) L. 265.

Fig 7. The legend is not easy to read, please make the line of the legend and the figure thicker.

Answer. Done (now Figure 6)

---

## [Editor Report · Decision Letter 1]

31 Jan 2020

Vegetation changes in temperate ombrotrophic peatlands over a 35 year period

PONE-D-19-25600R1

Dear Dr. Pellerin,

We are pleased to inform you that your manuscript has been judged scientifically suitable for publication and will be formally accepted for publication once it complies with all outstanding technical requirements.

With kind regards,

RunGuo Zang

Academic Editor

PLOS ONE

Additional Editor Comments (optional):

accept
---

## [Editor Report · Acceptance letter]

6 Feb 2020

PONE-D-19-25600R1 

Vegetation changes in temperate ombrotrophic peatlands over a 35 year period 

Dear Dr. Pellerin:

I am pleased to inform you that your manuscript has been deemed suitable for publication in PLOS ONE. Congratulations! Your manuscript is now with our production department. 

With kind regards,

on behalf of

Professor RunGuo Zang 

Academic Editor

PLOS ONE